# Dermatologic Changes in Experimental Model of Long COVID

**DOI:** 10.3390/microorganisms12020272

**Published:** 2024-01-27

**Authors:** Hussain Hussain, Michael J. Paidas, Ramamoorthy Rajalakshmi, Aya Fadel, Misha Ali, Pingping Chen, Arumugam R. Jayakumar

**Affiliations:** 1Department of Obstetrics, Gynecology and Reproductive Sciences, University of Miami Miller School of Medicine, Miami, FL 33136, USA; hussainhussainmd77@gmail.com (H.H.); rxr1310@med.miami.edu (R.R.); 2Department of Internal Medicine and Infectious Disease, Larkin Community Hospital, Miami, FL 33143, USA; 3Department of Biochemistry and Molecular Biology, University of Miami Miller School of Medicine, Miami, FL 33136, USA; 4Department of Internal Medicine, Ocean University Medical Center—Hackensack Meridian Health, Brick Township, NJ 08724, USA; ayafadel167@yahoo.com; 5Department of Radiation Oncology, Sylvester Comprehensive Cancer Center, University of Miami Miller School of Medicine, Miami, FL 33136, USA; mxa5252@med.miami.edu; 6Department of Pediatrics, University of Miami Miller School of Medicine, Miami, FL 33136, USA; pchen2@med.miami.edu

**Keywords:** dermatologic manifestation, fibrosis, inflammation, long COVID, murine hepatitis virus-1

## Abstract

The coronavirus disease-19 (COVID-19) pandemic, declared in early 2020, has left an indelible mark on global health, with over 7.0 million deaths and persistent challenges. While the pharmaceutical industry raced to develop vaccines, the emergence of mutant severe acute respiratory syndrome coronavirus-2 (SARS-CoV-2) strains continues to pose a significant threat. Beyond the immediate concerns, the long-term health repercussions of COVID-19 survivors are garnering attention, particularly due to documented cases of cardiovascular issues, liver dysfunction, pulmonary complications, kidney impairments, and notable neurocognitive deficits. Recent studies have delved into the pathophysiological changes in various organs following post-acute infection with murine hepatitis virus-1 (MHV-1), a coronavirus, in mice. One aspect that stands out is the impact on the skin, a previously underexplored facet of long-term COVID-19 effects. The research reveals significant cutaneous findings during both the acute and long-term phases post-MHV-1 infection, mirroring certain alterations observed in humans post-SARS-CoV-2 infection. In the acute stages, mice exhibited destruction of the epidermal layer, increased hair follicles, extensive collagen deposition in the dermal layer, and hyperplasticity of sebaceous glands. Moreover, the thinning of the panniculus carnosus and adventitial layer was noted, consistent with human studies. A long-term investigation revealed the absence of hair follicles, destruction of adipose tissues, and further damage to the epidermal layer. Remarkably, treatment with a synthetic peptide, SPIKENET (SPK), designed to prevent Spike glycoprotein-1 binding with host receptors and elicit a potent anti-inflammatory response, showed protection against MHV-1 infection. Precisely, SPK treatment restored hair follicle loss in MHV-1 infection, re-architected the epidermal and dermal layers, and successfully overhauled fatty tissue destruction. These promising findings underscore the potential of SPK as a therapeutic intervention to prevent long-term skin alterations initiated by SARS-CoV-2, providing a glimmer of hope in the battle against the lingering effects of the pandemic.

## 1. Introduction

The global impact of the COVID-19 pandemic persists, with millions affected and ongoing morbidity and mortality. As of 12 January 2023, deaths have exceeded 7.7 million since the pandemic declaration in early 2020, accompanied by a daily influx of over 39 thousand new cases worldwide [1]. On a positive note, 767 million individuals have successfully recovered from mild to severe SARS-CoV-2 infections out of 775 million affected individuals [1]. Despite concerted global efforts to develop vaccines aimed at reducing mortality, no specific treatment exists to date for the viral infection [1,2,3,4].

Beyond the acute complications of COVID-19, survivors grapple with enduring health issues, encompassing disrupted sleep patterns, osteoporosis, subfertility, exacerbated diabetes, fatigue, and complications affecting musculoskeletal, cardiovascular, gastrointestinal, pulmonary, neurologic, and urologic systems [1,2,5,6,7]. The intricate interplay of SARS-CoV-2 genetic mutations and the array of documented post-infection complications challenge scientists in pinpointing the exact pathophysiology. Furthermore, the progression of disease pathogenesis, spanning weeks to months, reveals signs of multiple organ dysfunction or failure occurring in one or more episodes of post-acute SARS-CoV-2 infection—referred to as post-acute sequelae of SARS-CoV-2 infection with or without overt symptoms and behavioral changes.

Investigations have delved into a clinical syndrome exhibiting similarities to severe acute respiratory syndrome (SARS) in mice infected with murine hepatitis virus-1 (MHV-1). This syndrome is distinguished by a markedly high mortality rate [8,9,10,11,12,13]. These mice exhibit pronounced lung injury with mortality rates ranging from 40% to 60% between days 7 and 12 post-infection [14]. Upon examination at the time of death, the lungs display severe interstitial pneumonitis characterized by interstitial inflammatory reactions, along with substantial infiltrations of lymphocytes and macrophages [15,16,17,18]. Additionally, investigations into the livers of MHV-1-infected A/J mice disclose evidence of severe hepatic congestion, mirroring observations in humans infected with SARS-CoV-2 [16,18].

In addition to multi-organ dysfunction, skin changes associated with COVID-19 have been established in acute stages post-infection, which encompass a range of manifestations that can vary widely among individuals. One common type is the maculopapular rash, characterized by red macules, and papules. In some cases, individuals with COVID-19 exhibit a vesicular rash, consisting of fluid-filled blisters that can resemble those seen in chickenpox or herpes infections [19,20]. However, this presentation is less common compared to other skin appearances. COVID toes are another distinctive skin change, involving red or purple discoloration of the toes and fingers, often accompanied by swelling and pain [20]. This phenomenon is more frequently observed in younger individuals afflicted with COVID-19 and has been associated with milder cases of illness. Urticaria, or hives, is characterized by raised, itchy welts on the skin and can occur independently or alongside other COVID-19 symptoms [21,22]. Petechiae and purpura, small red or purple spots, and larger discolorations, respectively, may also be present, particularly in severe cases due to coagulation abnormalities associated with the virus or vasculitis [22,23,24,25,26,27,28]. In some cases, a mottled, purplish discoloration known as livedo reticularis was developed, thus forming a net-like pattern on the skin [25]. This is often associated with changes in blood flow and is observed in severe cases, especially in individuals with pre-existing vascular issues [25,28].

Given the common genus shared by MHV severe acute respiratory syndrome coronavirus and SARS-CoV-2, insights gleaned from MHV-1 could potentially provide a mechanistic understanding of SARS-CoV-2 infection in humans [14]. While there are notable similarities between MHV-1 in mice and SARS-CoV-2 in humans, such as certain pathogenic features, differences exist (e.g., there are variations in viral binding receptors angiotensin-converting enzyme-2 (ACE2) in SARS-CoV-2 versus carcinoembryonic antigen-related cell adhesion molecule 1 (CEACAM1) in MHV-1) [15,16,18,29]. Distinctions include the proteolytic cleavage of four crucial amino acids at the S1/S2 site of the SARS-CoV-2 spike protein [29]. Despite these variances, the observed similarities outweigh the differences. Importantly, our identification of pathological and functional changes in the MHV-1 mice model of COVID-19 demonstrates a high degree of comparability to humans afflicted with SARS-CoV-2 infection [15,16,18,29].

In this study, we aimed to investigate the association between acute and long-term dermatological changes in COVID-19. We further examined whether the inhibition of viral entry by a newly identified 15-amino-acid synthetic peptide, SPIKENET (SPK, which was effective in preventing Spike glycoprotein-1 binding with host receptors, and has a potent anti-inflammatory effect on severe inflammatory stimuli), prevents or ameliorates skin alterations.

## 2. Materials and Methods

### 2.1. Mice

We used 8-week-old female A/J mice weighing 22–24 g each. These mice were purchased from Jackson Laboratories (Bar Harbor, ME, USA), and were kept in cages at the University of Miami Miller School of Medicine animal isolation facility. The animals were fed with a standard lab chow diet (Envigo 2918 irradiated, Teklad diet, Dublin, VA, USA) and provided with water ad libitum (autoclaved tap water). The study was performed according to the guidelines of the University of Miami Institutional Animal Care and Use Committee (IACUC protocol number 20-131 LF/Renewed protocol number, 20-162). In the acute investigation, these mice were divided into 3 groups: MHV-1 infection alone (n = 16), healthy control (n = 7), and SPK treated mice (n = 5). For the long COVID study, we investigated a total of 12 mice (4 MHV-1 infection, 4 healthy control, 4 SPK treated group).

### 2.2. Viral (MHV-1) Inoculation and SPIKENET Treatment

MHV-1 was purchased from the American Type Culture Collection (ATCC, cat# VR 261, Manassas, VA, USA). Mice were split into (1) a healthy control group, (2) an MHV-1 virus-inoculated group, and (3) an MHV-1 virus-inoculated group that was treated with SPK. MHV-1 viral inoculation was performed as previously described [14,15,16,18]. Briefly, groups 2 and 3 were inoculated intranasally with 5000 PFU MHV-1 and were observed to ensure adequate inhalation of the virus. MHV-1 inoculated mice (group 3) were additionally treated with 5 mg/kg body weight of SPIKENET (SPK) as previously described by us [14,15,16,18].

### 2.3. Skin Collection and Storage

The skin of the mice was collected, and hair was removed and placed in 10% formalin (7 days post-infection for acute studies and 12 months post-infection for long-term studies). Paraffin embedded sections (processed through Histoscore PELORIS 3 Premium Tissue Processing System, Leica Biosystems Inc., Buffalo Grove, IL, USA) were then cut into 10 µm thick pieces by an ultra-thin semiautomatic microtome (Histoscore auto cut automated rotary microtome, Leica Biosystems Inc., Buffalo Grove, IL, USA).

### 2.4. Histological Staining

Histological staining of the mice skin with hematoxylin and eosin (H&E) was performed using the following materials and reagents; formalin-fixed paraffin-embedded mice skin tissue sections, xylene, absolute ethanol, 95% ethanol, 70% ethanol, hematoxylin solution, eosin Y solution, distilled water, microscope slides, and cover slips [30]. Initially, deparaffinization was performed through the following steps: the slides were placed in xylene for 5 min followed by absolute ethanol for 3 min, immersed in 95% ethanol for 2 min, and then placed in 70% ethanol for 2 min [30]. The tissue sections were rinsed with distilled water. The hematoxylin staining was performed by adding the slides in hematoxylin solution for 5–10 min, then rinsing with distilled water to remove excess stain, and differentiating the slides with 1% acid alcohol until sections turned blue. The slides were then rinsed again in distilled water [30]. Eosin staining was established by exposing slides to eosin Y solution for 2–3 min and then rinsing with distilled water [30]. The slides underwent a dehydration step through adding them in 70%, 95%, and absolute ethanol; then, the slides were placed in xylene for 5 min. The slides were then mounted in mounting media and the cover slips were applied [30].

### 2.5. Immunofluorescence

Paraffin-embedded tissue sections from MHV-1 infected groups (acute and long COVID), measuring 10 microns, were incubated with antibodies specific to viral proteins as described previously by us [16]. SARS-CoV-2 (COVID-19) Spike S1 rabbit monoclonal antibody (Cat# GTX635671; GeneTex, Irvine, CA, USA) and SARS-CoV-2 Nucleocapsid mouse monoclonal antibody (Cat# MA5-29981: Invitrogen, Waltham, MA, USA) were used at 1:200 dilution. Horseradish peroxidase-conjugated anti-rabbit and anti-mouse secondary antibodies (Vector Laboratories, Newark, CA, USA) were used at 1:500 dilution. Immunofluorescent images were acquired with a Zeiss LSM510/UV Axiovert 200 M confocal microscope (Carl Zeiss Microscopy, LLC, Thornwood, NY, USA) with a plan apochromat ×40 objective lens and ×2 zoom, resulting in images of 125 × 125 μm in the area and 1.0 μm optical slice thickness (1.0 Airy units for Alexa Fluor 546 or 568 emission channel). The images were randomly collected in a “blinded” manner. At least 12 fluorescent images were captured per mouse and merged to colocalize with Spike S1 and Nucleocapsid.

### 2.6. Quantitation of Skin Thickness

Hematoxylin and eosin (H&E)-stained slides from both acute and long-term post-infection were acquired using a microscope (Olympus VS120 Automated Slide Scanner, Olympus, Pittsburgh, PA, USA) and were measured for thickness using an online tape. Briefly, the images obtained (23×) were enlarged to 3×, and the thickness of various regions (epidermis, dermis, and adipose tissue) was obtained using online tape and the thickness was expressed as millimeters. GraphPad Prism (Version 10.1.2, GraphPad Software, Boston, MA, USA) was used to analyze the data with Tukey’s multiple comparison test.

### 2.7. Statistical Analysis

Data were subjected to analysis of variance followed by Tukey’s multiple comparison test. A statistical analysis showing *p* < 0.05 was considered significant.

## 3. Results

Diverse histological alterations were discerned during our scrutiny of MHV-1 infection within the murine integument, probing into the acute, long COVID, and SPK treatments, as well as SPK alone (SPK was not displaying any changes in the control group, which was identical to the healthy group, figure is not shown). In the mature murine cohort, the epidermal stratum exhibited a characteristic semblance, showcasing the canonical quartet of epithelial-derived cell strata identified in analogous taxa: the stratum basale, stratum spinosum, stratum granulosum, and the keratinized stratum corneum. The stratum basale, accommodating actively proliferating cells, emerged as the exclusive stratum where mitotic activity attained visibility in the unblemished cutaneous tissue. It was noteworthy that apparent basophilic granules were identified within the stratum granulosum. Further, a melanin pigmentation manifested within the cells of the stratum basale, as captured in Figure 1A,C.

The cutaneous histopathology displays a pronouncedly compromised epidermal stratum characterized by a keratosis, papillomatosis, and instances of apoptotic events among the epidermal cells. Furthermore, we found conspicuous features encompassing hyperplasticity in sebaceous gland development and a pronounced deposition of collagens within the lower epidermal and dermal strata. A discernible augmentation in the number of anagenic hair follicles is observed, concomitant with heightened sebaceous gland development. It is noteworthy that the attenuation of the panniculus carnosus and adventitial layer, coupled with dermal thickening attributable to the extensive destruction/obliteration of the fatty tissue layer and architectural modification throughout the skin strata, were observed (Figure 1B,D and Figure 2D). Moreover, we detected a complete abolition of select epidermal regions (Figure 2B), as compared to the control mice that were not inoculated with the virus (Figure 2A). Additionally, the presence of bullae dispersed within the dermal layer of the infected mice was detected, as illustrated in Figure 2D; this was in stark contrast to the normative cutaneous morphology identified in non-infected mice, depicted in Figure 2C.

We next examined potential cutaneous alterations in the long COVID phase, precisely 12 months post-MHV-1 infection. The histopathological paradigm is illustrated in Figure 3B, showing pronounced architectural perturbation, extensive attenuation of the epidermal stratum, keratinocyte loss, diverse apoptotic bodies, and the conspicuous absence of hair follicles within the dermal stratum (Figure 3B and Figure 4B). A comparative analysis with the control group reveals a discernible thinning of the panniculus carnosus in the long COVID cohort (Figure 3A and Figure 4A). Concurrently, we observed adipose tissue degeneration/fibrosis concomitant with the obliteration of the adventitial layer (Figure 3B); these are phenomena reminiscent of those observed during the acute phase infection.

Remarkably, the treatment of the MHV-1-infected mice with the synthetic peptide SPK led to a substantial amelioration of the cutaneous changes discerned in long COVID. Notably, SPK intervention effectively reinstated the morphogenesis of hair follicles (Figure 3D) in stark contrast to the untreated control group (Figure 3C). Fascinatingly, this therapeutic intervention demonstrated the restoration of architectural integrity not only in the adipose tissue but also across the dermal and epidermal strata, encompassing the sebaceous glands (Figure 4D) juxtaposed to the untreated control mice (Figure 4C). Additionally, the sebaceous glands exhibit conspicuous dispersion along the length of hair follicles, accompanied by a discernible augmentation in the thickening of the panniculus carnosus (Figure 3D), which is notably absent in the untreated control group (Figure 3C).

In Figure 5B,C, diverse stages of hair follicle degeneration are evident long-term post-infection, as juxtaposed with the normal, healthy mice illustrated in Figure 5A. The application of SPK in infected mice led to the restoration of hair follicles across distinct stages, along with the regeneration of adipose tissue. Figure 6, portraying acutely infected mice (A–D), reveals microthrombosis (indicated by black and brown arrows in A and B, respectively), dermal edema (depicted by the blue arrow in A), keratinocyte apoptosis (highlighted by the green arrow in B), apoptotic cells surrounding a dying hair follicle encircled by edema, and Merkel cells surrounded by edema (marked by blue and black arrows in C, respectively). Furthermore, in D, observations included edema around the hair follicles (brown arrow), a group of melanocytes (blue arrow), some of which are undergoing degeneration, and intra-keratinocyte vacuolization. In Figure 7, notable alterations were observed. Figure 7A exhibits the enlargement of sebaceous cells with intracellular vacuolization (denoted by the black arrow), while the red arrow indicates the presence of an eosinophil. Figure 7B depicts bullae, and Figure 7D reveals a nest of inflammation (indicated by the blue arrow), and thrombotic vessels (marked by the black arrow), with the red arrow highlighting melanocyte intracellular edema. Conversely, the green arrow points to the presence of edema. According to the changes above, we quantitated alterations in skin thickness across acute, long COVID, and treatment groups (Figure 8 and Figure 9).

In the acute phase of the infection, we have reported, for the first time, the dissemination of coronavirus through the bloodstream to various organs, including the skin. This builds upon previously documented findings of viral load detection in the blood and the presence of the virus within multiple organs [14,15,16,17,18,31,32]. We also reported the presence of the virus and viral particles in the brains of infants born to mothers infected with SARS-CoV-2 [31]. Furthermore, we now show the localization of viral particles within the skin in both the acute and long COVID groups (Figure 10). Indeed, viral particles, specifically nucleocapsids, have been identified within the nuclear membrane. Viral particles were also sequestrated in the nucleus, which suggests possible adverse effects in both acute and long-term post-infection, as studies have showed cell injury when exposed to SARA-CoV-2 viral particles in vitro [33,34,35]. In addition to their presence in the nuclear environment, the virus has also been observed in the cytoplasm (Figure 10 right side B). Notably, the presence of the virus was not detected in the skin long-term post-infection (Figure 10).

## 4. Discussion

The histopathological changes in acute and long-term post-COVID patients have not been investigated so far. Several cutaneous abnormalities were detected during our investigation in a mice model of acute and long COVID. In the acute phase of the disease, we observed the destruction of the epidermal layer, an increase in the number of hair follicles, an extensive deposition of collagen in the dermal layer, and the hyperplasticity of the sebaceous glands. Further, the investigation disclosed the thinning of the panniculus carnosus and the adventitial layer. In contrast, the long-term post-COVID cutaneous investigation showed the absence of hair follicles from the epidermal and dermal layers, destruction of adipose tissues, and obliteration of the epidermal layer. Further, the treatment of infected mice with SPK reversed these cutaneous abnormalities. Precisely, the drug was able to restore the number of hair follicles and re-architecture the epidermal and dermal layers. Moreover, adipose tissue was restored in the treated mice, which strongly suggests the possibility of long-term cutaneous defects in humans with COVID-19; SPK can be used as a potential therapeutic agent.

Since the onset of the COVID-19 pandemic in 2020, a comprehensive examination of its clinical manifestations has unveiled diverse and often enigmatic multi-organ pathological changes and complications. In our prior study, we documented diverse clinical manifestations and alterations in mice during both the acute and prolonged phases of COVID-19 [15]. Our investigations have particularly focused on elucidating the potential alterations in the skin, revealing a spectrum of pathological changes in a well-established mouse model of COVID-19. Over time, it is anticipated that various dermatological disorders may emerge as sequelae of COVID-19. These potential disorders encompass but are not limited to alopecia, vitiligo, skin malignancy, accelerated skin wrinkling, pemphigus, and atopic dermatitis, among others. The etiology of these dermatological manifestations is likely multifactorial, involving intricate interactions between the virus and the host’s genes. While the virus itself may act as a potential trigger, its role in directly contributing to genetic mutations that lead to these disorders remains an area of ongoing investigation. It is essential to recognize the complexity of the SARS-CoV-2 virus and its potential mechanistic involvement in the development of dermatological disorders.

The International League of Dermatological Societies (ILDS), in collaboration with the American Academy of Dermatology (AAD), has meticulously documented a compendium of 2500 instances emanating from diverse regions, totaling up to 72 countries [36]. Predominant dermatological manifestations correlated with COVID-19 were identified. These include the following manifestations: (1) exanthematous rash (20%), (2) pernio-like acral lesions (19%), (3) urticaria (15%), (4) varicella-like eruption (12%), (5) papulosquamous rash (10%), and (6) retiform purpura (7%) [37,38]. Additionally, sporadic occurrences of multisystem inflammatory syndrome in children (MIS-C) have been ascribed to COVID-19 infections, with documented instances in the scientific literature [37,38]. It is noteworthy that MIS-C exhibits similarities to Kawasaki Disease; however, it is differentiated by distinctive clinical characteristics, encompassing the mean age of onset, cardiovascular sequelae, and pertinent laboratory parameters [37,38,39].

During the acute phase of MHV-1 infection, a conspicuous augmentation in the numerical density of hair follicles was distinctly observed in comparison to the control group. This phenomenon can be explicated by reference to the established pathophysiological mechanisms operative in non-COVID-19 cases. The principal factors contributing to this phenomenon encompass heightened androgen receptor expression and elevated levels of insulin-like growth factor-1 (IGF-1) [40,41,42,43]. Androgenic influence assumes a pivotal role in this context, acting as an inhibitory hormone within the scalp, while exerting a stimulatory effect in other anatomical regions [42]. Remarkable is the indispensable involvement of androgen-metabolizing enzymes, namely aromatase, 17 β-hydroxysteroid dehydrogenase (17 β-HSD), and 5-alpha reductase (types I and II), in the intricate orchestration of hair follicle development [42,43]. Concurrently, an array of hormones actively participate in fostering the developmental trajectory of hair follicles. These include estradiol, progesterone, prolactin, thyroid hormones (T3 and T4), melatonin, corticotropin-releasing hormone (CRH), adrenocorticotropic hormone (ACTH), cortisol, and thyroid-releasing hormone [40,41,42,43,44,45].

However, there exist inhibitory hormones that exert a regulatory influence on hair follicle development, such as galanin (predominantly localized in the peripheral and central nervous systems, with documented effects including the reduction of matric keratinocyte proliferation, shortening of the anagen phase, and attenuation of hair shaft extension) and transforming growth factor-β (TGF-β) [46,47,48].

In the prolonged course of COVID-19, a marked and complete abatement of hair follicle numbers was identified, which was characterized by severe obliteration and diminished follicular presence. This deleterious phenomenon is thought to be attributable to multiple factors, including the direct impact of the virus, the influence of TGF-β [48], and alterations in the hormonal milieu [49]. Notably, our experimental intervention involving the utilization of our proprietary therapeutic agent (SPK) yielded a remarkable resurgence in hair follicle numbers. This promising outcome suggests the potential efficacy of SPK in ameliorating or reversing alopecia induced by COVID-19. Consequently, this innovative medication holds promise as a prospective therapeutic modality for the treatment of COVID-19-associated alopecia in the future.

Diverse viruses exhibit an adept capacity to infiltrate the integumentary system, specifically establishing residence within the basal layer of the epidermis. Among the notable viral entities engaging in this modus operandi are herpes simplex, vaccinia virus, molluscum contagiosum, varicella-zoster, human papillomavirus, polyomavirus, human immunodeficiency virus (HIV), Zika virus, West Nile virus, Dengue virus, Chikungunya virus, and an array of other pathogens [50,51,52,53,54]. These viruses strategically exploit the epidermal stratum as a conducive environment for their replication processes [50,51,52,53,54], with keratinocytes emerging as the principal cellular cohort for such proliferative endeavors [50,51,52,53,54]. The epidermis, a proliferative substrate, becomes the locus for the intricate orchestration of viral replication, thereby facilitating the propagation and dissemination of these viral agents [54,55,56].

Shortly following entry, the toll-like receptor-2 (TLR) is promptly engaged, initiating a cascade that amplifies the activation of the nuclear factor-kappa B (NF-κB) pathway [55]. This orchestrated activation event precipitates the induction of an array of pro-inflammatory cytokines, notably interleukins (IL-1, IL-6, and IL-12), tumor necrotic factor-alpha (TNF-alpha), and chemokines, such as chemokine ligand-2 [55,56]. Concomitantly, the production of additional inflammatory constituents, including interferons and histamine, ensues, thereby fostering the recruitment of inflammatory cells to the affected site [56]. This heightened recruitment of inflammatory cells, while constituting an immune response, paradoxically begets consequential deleterious effects. The ensuing influx of inflammatory cells inflicts severe and undesirable injury upon the cellular constituents of the epidermal layer [54,56]. This injurious process is observable in both acute and long-term manifestations of COVID-19, wherein a conspicuous and consequential detriment to the structural integrity of the epidermal layer is evident. The observed severe damage underscores the complex interplay between viral-induced immune responses and the resultant pathological consequences within the epidermal microenvironment.

Substantial evidence has been amassed, attesting to the discernible presence of SARS-CoV-2 within the skin, substantiated by the observable loss of adipose tissue during both acute and protracted phases of infection. Notably, the adipose tissue undergoes a conspicuous and seemingly complete disappearance, as delineated above. This peculiar manifestation is ascribed to the virus’s proclivity for extrapulmonary adipose tissue targeting, as Thangavel et al. elucidated in a murine model [57]. The investigative work by Thangavel et al. not only underscored the virus’s fondness for adipose tissue but also shed light on the infiltration of inflammatory cells, particularly macrophages, within the afflicted adipose tissue [57].

In light of these revelatory findings, our research endeavors were extended to ascertain the potential therapeutic effects of SPK. Strikingly, as illustrated in Figure 4 above, the outcomes of our investigations revealed a noteworthy restoration of both the epidermal layer and the adipose tissue layer following SPK administration. This unforeseen recuperation implies a promising role for SPK in mitigating the deleterious effects on both cutaneous components inflicted by SARS-CoV-2 infection, suggesting a potential avenue for therapeutic intervention in the context of skin manifestations associated with COVID-19.

The notable deposition of collagen emerges because of the inflammatory processes induced by the viral infection [58]. Within the intricate milieu of the extracellular matrix (ECM), a myriad of constituent elements contributes to its structural complexity. These include various types of collagens (I–XXIII), elastic fibers (comprising elastin, fibrillin-1, and fibrillin-2), glycoproteins (such as laminin, fibronectin, nidogen, entactin, periostin, tenascin-C, and thrombospondin-1), proteoglycans (encompassing biglycan, decorin, perlecan, versican, lumican, fibromodulin, and agrin), galactosaminoglycan, and glycosaminoglycan (including heparan sulfate, dermatan sulfate, and hyaluronan) [59,60,61,62,63,64,65,66,67,68,69,70]. This comprehensive array of ECM components underscores the intricate and dynamic nature of the tissue microenvironment, where the interplay of diverse molecular constituents contributes to the structural integrity and functionality of tissues. The observed collagen deposition, linked to the viral-induced inflammatory response, underscores the complexity of the molecular landscape orchestrating tissue remodeling and repair in the context of viral infections.

ECM stands as a perpetually modulated, evolutionarily conserved, highly adaptable, and dynamically transformative scaffold, orchestrating controlled processes of metamorphosis and restructuring [67]. Within the integumentary system, the skin’s layers, comprising the epidermis and dermis, are demarcated by the basement membrane. This intricate amalgamation of ECM proteins and fibers assumes the pivotal role of bridging the epithelium to the subjacent tissue, thereby establishing a structural and functional continuum [67,68,69]. Within the interstitial expanse of the dermis, a diverse array of cells, including fibroblasts and immune/vascular cells, reside proximate to the ECM, providing a supportive milieu and ensconcing these cells in a manner conducive to intimate interactions [69,70]. This dynamic cellular–ECM interplay contributes to the maintenance of tissue homeostasis and function. However, any aberrations in the ECM, particularly those induced by pathological agents such as bacterial or viral infections, can exert discernible effects on the dermal layer. Such alterations manifest through the recruitment of various immune cells, thereby instigating a cascade of events that influence the structural and physiological aspects of the dermal microenvironment.

In our experimental model, a conspicuous augmentation in the abundance of the ECM was noted, with discernible evidence of heightened collagen deposition observable in Figure 1 and Figure 2. This substantial increase in the ECM can be ascribed to incremental alterations occurring in both fibroblasts and keratinocytes, which emerge as principal catalysts driving the remodeling processes within the ECM. Notably, our earlier investigations have revealed a significant elevation of TGF-β within the renal tissue [32]. TGF-β, conventionally latent in the extracellular matrix, undergoes activation in response to various stimuli, including tissue injury or inflammatory cues [32,71]. Upon activation, TGF-β engages with its cell surface receptors, predominantly serine/threonine kinase receptors, recognized as TGF-β receptors [71]. This activation of TGF-β and the subsequent interaction with its receptors constitute pivotal molecular events contributing to the orchestrated modulation of the ECM [72]. The intricate interplay between these cellular and molecular constituents underscores the multifaceted nature of ECM dynamics and the regulatory mechanisms governing tissue remodeling in response to various stimuli.

TGF-β initiates an intracellular signaling cascade, involving proteins like Smad proteins that transduce the TGF-β signal from the cell membrane to the nucleus [73,74,75,76]. In the nucleus, Smad proteins, along with other transcription factors, regulate the expression of genes involved in collagen synthesis, including genes that encode various types of collagens, the principal component of the extracellular matrix [73,75]. As a result of these signaling pathways, fibroblasts, the primary cells responsible for collagen production, are stimulated to synthesize and release increased amounts of collagen [73,75]. The excess collagen is then deposited in the extracellular matrix, leading to remodeling and alterations in the structure of the skin.

Consequently, it is plausible that the virus can elicit an augmented secretion of TGF-β, either directly or indirectly, by activating periostin in fibroblasts and basal keratinocytes during both the acute and protracted phases of COVID-19 [32,75]. Remarkably, our investigations have delved into the impact of SPK on skin tissue remodeling, revealing discernible outcomes such as a reduction in collagen accumulation and fibrosis within the dermal layer. Noteworthy is the consistent observation that TGF-β levels were attenuated in our prior renal model upon employing SPK [32]. This confluence of findings suggests a potential modulatory effect of SPK on the TGF-β signaling pathway, mitigating its downstream effects on collagen deposition and fibrosis. The multifaceted influence of SPK on distinct tissue types underscores its therapeutic potential in attenuating pathological alterations associated with COVID-19, indicating that SPK is a promising candidate for interventions aimed at mitigating tissue remodeling and fibrosis in diverse anatomical contexts.

## 5. Conclusions

The study found significant skin changes in mice during acute and long-term phases after MHV-1 infection. Epidermal destruction, increased hair follicles, extensive collagen deposition, and sebaceous gland hyperplasticity were observed in mice 7 days post-infection (acute stage). Long-term effects (one-year post-infection) include loss of hair follicles, adipose tissue destruction, and additional epidermal damage. The treatment of the MHV-1 infected mice with the synthetic peptide SPIKENET (SPK) successfully restored skin features in infected mice. These results suggest possible acute and long-term skin manifestation in humans with long COVID, and that SPK can be a potential therapeutic intervention for preventing long-term skin alterations caused by SARS-CoV-2, offering hope in addressing pandemic-related effects.

*Future perspectives*: It should be noted that the above-mentioned skin changes and possible mechanisms in the discussion section, as well as our current observations, strongly suggest the probable sensitization of the skin long-term post-infection that may enhance future severe skin defects when exposed to other types of infections or even with recurrent COVID-19. Therefore, further research is warranted regarding the long-term cutaneous changes and mechanisms involved in such skin defects that could assist in developing novel therapies to ameliorate the skin disease progression associated with COVID-19. 

## Figures and Tables

**Figure 1 microorganisms-12-00272-f001:**
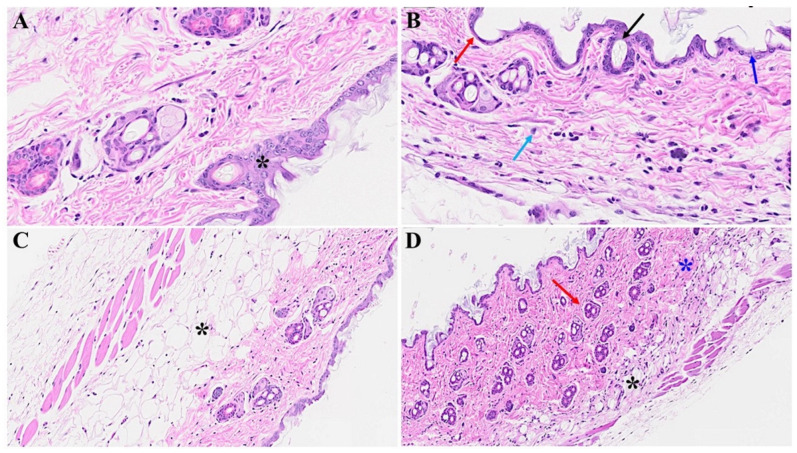
Acute cutaneous changes in mice infected with MHV-1 (7 days post-infection). There is thinning of the epidermis (red arrow in (**B**)) with the absence of stratum corneum (dark blue arrow), papillomatosis (black arrow), and dying keratinocyte (light blue arrow) in MHV-1 infected mice. In contrast, the epidermal layer is preserved in healthy mice ((**A**), asterisk). The adipose tissue is replaced by fibrosis and collagen deposition in MHV-1 infected mice (black asterisk in (**D**)), as compared to the control ((**C**), asterisk). Additionally, extensive collagen deposition is observed in MHV-1-infected mice ((**D**), blue asterisk), and the number of hair follicles is increased, as compared to the controls (**A**,**C**). (**A**,**B**) = 66×; (**C**,**D**) = 23× in Figure 1 and Figure 2 below, (n = 16).

**Figure 2 microorganisms-12-00272-f002:**
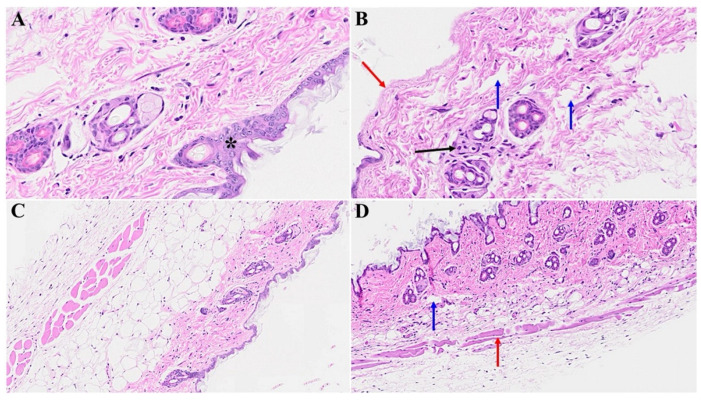
Acute skin alteration in MHV-1 infection (7 days post-infection). There is an area of absence of epidermis noted in MHV-1-infected mice (red arrow in (**B**)), as compared to controls (asterisk in (**A**)). Additionally, there are various forms of dermal edema (blue arrows in (**B**)), and hyperplasticity of the sebaceous gland (black arrow in (**B**)). The normal skin architecture is observed in (**C**), while, there are various bullae and thinning of panniculus carnosus in MHV-1-infected mice (blue and red arrows, (**D**), respectively), (n = 16).

**Figure 3 microorganisms-12-00272-f003:**
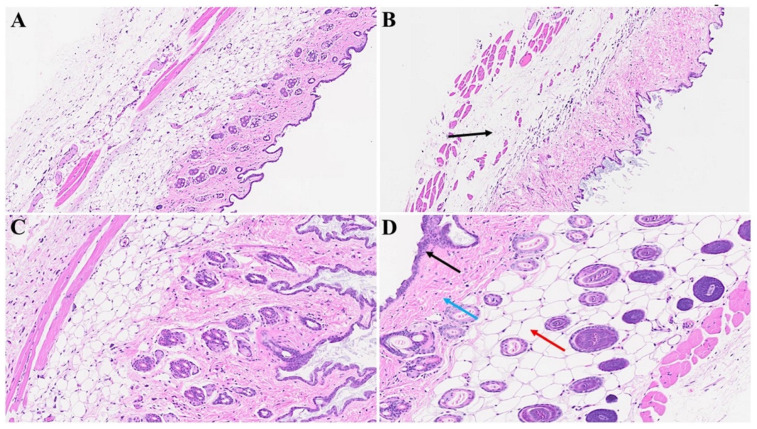
Skin changes long-term post-MHV-1 infection in mice (1-year post-infection). There is a destruction of the adipose tissue in MHV-1-infected mice (black arrow), and an absence of the hair follicles in a dermal layer in mice 1-year post-infection (**B**), as compared to healthy control mice (**A**). Treatment of MHV-1-infected mice with SPK (5 mg/kg, as previously described by us, [15]) reversed the hair follicle loss (**D**). The destruction of the epidermal layer, dermal, and adipose tissue was also restored using SPK (black, blue, and red arrows in (**D**)), respectively. (**A**,**B**) = 23×; (**C**,**D**) = 66× in Figure 3 and Figure 4 below, (n = 6).

**Figure 4 microorganisms-12-00272-f004:**
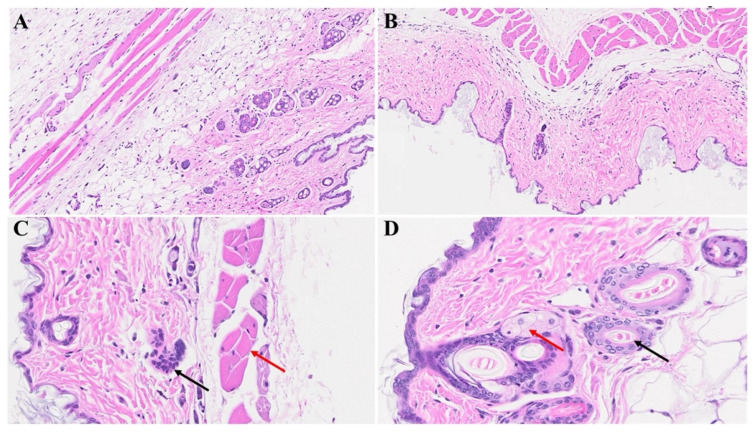
Cutaneous abnormalities in long COVID. We found extensive destruction of the epidermis, dermal, and adipose tissue in MHV-1-infected mice (1-year post-infection, (**B**)). We also identified severe dermal edema and demolition of the skin architecture (**B**), as well as dying hair follicles (black arrow in (**C**)), and distortion of the panniculus carnosus architecture (red arrow, in (**C**)) in these mice, as compared to the healthy controls (**A**). These cutaneous abnormalities were restored when MHV-1-infected mice were treated with 5 mg/kg SPK (**D**), (n = 6).

**Figure 5 microorganisms-12-00272-f005:**
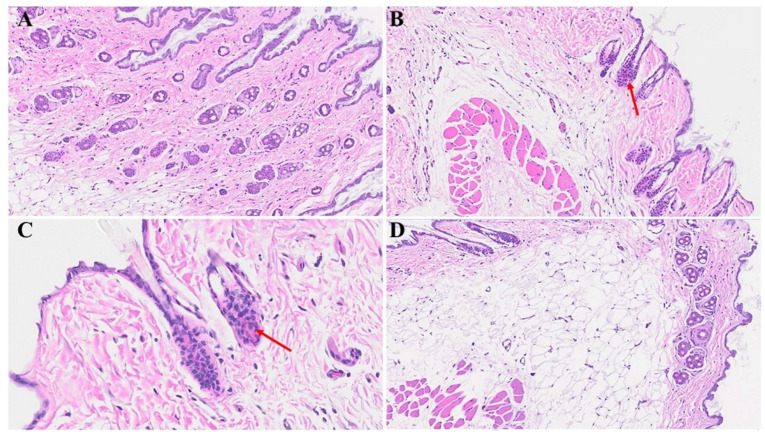
Skin changes long-term post-MHV-1 infection in mice. Dying hair follicles with surrounding inflammatory cells (red arrows in (**B**,**C**)), as well as the destruction of skin architecture were observed 1-year post-MHV-1 infection, as compared to controls (**A**). These skin aberrations were, however, not observed in MHV-1-infected mice that were treated with SPK (5 mg/kg) (**D**). (**A**,**B**,**D**) = 23×, and (**C**) = 66×, (n = 6).

**Figure 6 microorganisms-12-00272-f006:**
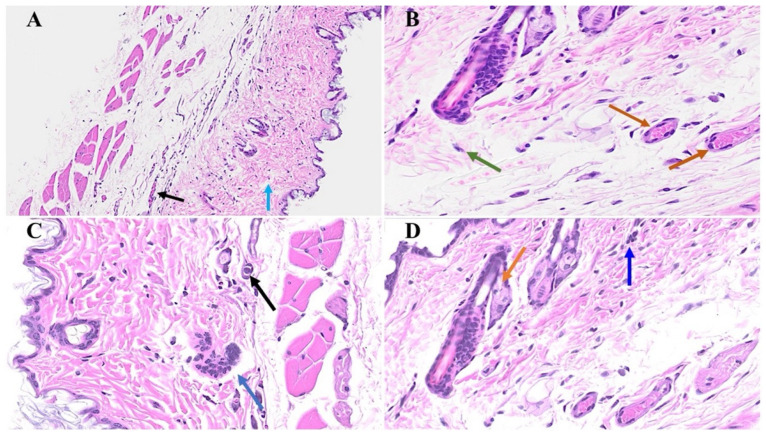
We noticed microthrombosis (black arrow in (**A**)), and various dermal edema (blue arrow in (**A**)). There was an apoptotic keratinocyte (green arrow) and microthrombi (brown arrows) 1-year post-MHV-1 infection (**B**). We also detected Merkel cells with mild intracellular edema, dying hair follicle encircled by edema (black and blue arrows in (**C**), respectively), as well as intra-sebaceous gland vacuolization (brown arrow in (**D**)), and dying melanocytes (blue arrow in (**D**)) in MHV-1-infected mice 1-year post-infection. (**A**) = 23×; (**B**–**D**) = 66×, (n = 6).

**Figure 7 microorganisms-12-00272-f007:**
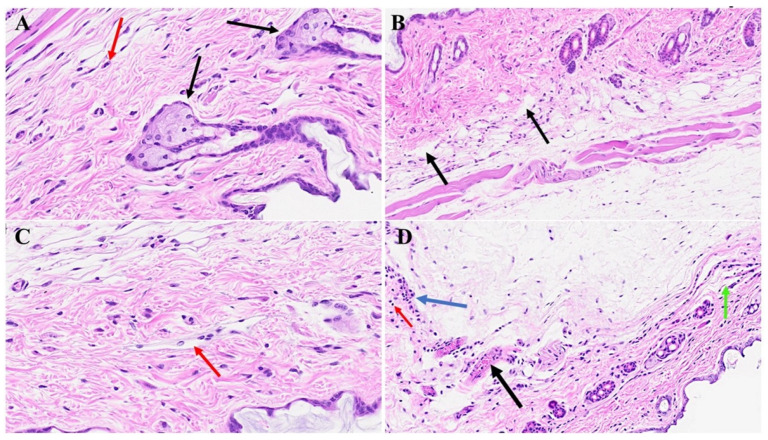
Skin changes long-term post-MHV-1 infection in mice. MHV-1-infected mice (1-year post-infection) show the presence of eosinophil (red arrow in (**A**)) and hyperplasia of sebaceous glands (black arrows in (**A**)). There is also dermal bulla filled with inflammatory cells (red arrow), as well as scattered inflammatory cells, along with dermal edema in these mice (**C**). We also found a nest of inflammation (blue arrow), edematous melanocyte (red arrow), thrombosis (black arrow), and intradermal edema (green arrow) (**D**) in MHV-1-infected mice (**A**–**D**) = 23×, (n = 6).

**Figure 8 microorganisms-12-00272-f008:**
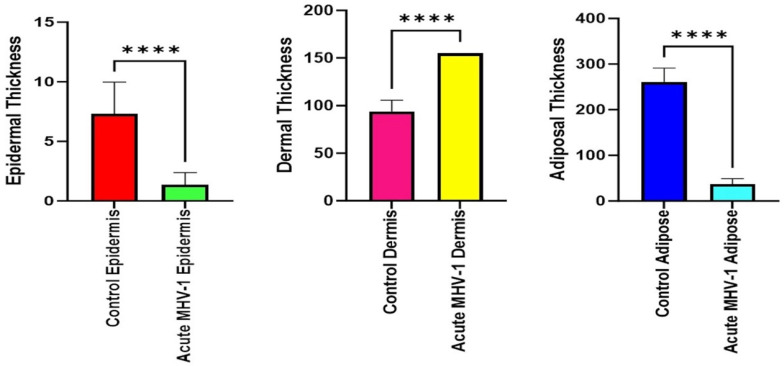
Quantitation of cutaneous thickness in an acute model revealed significant reductions in the epidermal and adipose tissue layers compared to the healthy group, while the dermal layer showed a significant increase. **** *p* < 0.0001 (statistically significant different from compared groups). Dermal thickness is presented in millimeters with 3× enlargement from original magnification (23×).

**Figure 9 microorganisms-12-00272-f009:**
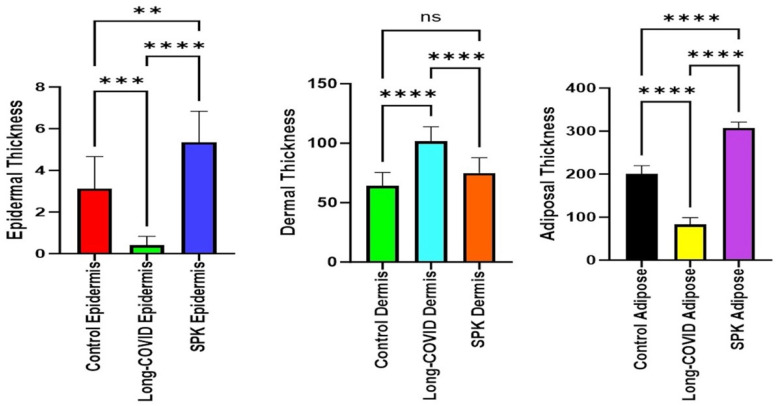
Quantitation of skin layer thickness alteration in the long COVID model showed a decrease in the epidermal and adipose tissue layers compared to the control mice. In contrast, the dermal layer displayed a significant increase. The SPK groups exhibited an increase in the thickness of the epidermal, dermal, and adipose layers. ** *p* < 0.05 (statistically significant different from compared groups). *** and **** *p* < 0.0001 (statistically significant from compared groups). ns = nonsignificant. Dermal thickness is presented in millimeters with 3× enlargement from original magnification (23×).

**Figure 10 microorganisms-12-00272-f010:**
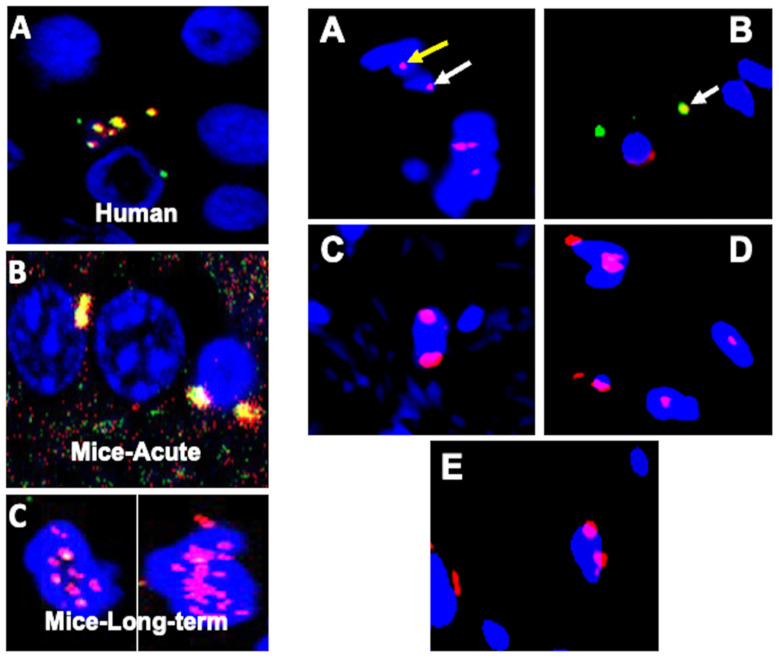
Identification of virus, and viral particles using immunofluorescence. Figure have two classifications: (**left side**) (**A**–**C**): (**A**) We earlier reported the presence of SARS-CoV-2 (orange color)-merged images of spike glycoprotein 1 (S1, green) and nucleocapsid protein (red) in the brain of an infant (human-one year), whose mother had SARS-CoV-2 infection during birth. (**B**) We now show the presence of the virus in mice brains 7 days post-MHV-1 infection. (**C**) Remarkably, we still observe viral particles (nucleocapsid protein, red) in the brains of MHV-1-infected mice (one-year post-infection), and that the viral particles are sequestered in the nucleus as opposed to their localization in the nuclear membrane in acute infection (**B**). (**Right side**) (**A**–**E**): Interestingly, nucleocapsid protein was detected in the nuclear membrane (white arrow, (**A**)) and viral particles in the nucleus (yellow arrow in (**A**)). This finding signifies the sequestration of viral particles in the skin nucleus. In addition to the nuclear membrane and nuclear sequestration, we also found the virus in the cytoplasm (white arrow in (**B**)). However, we only found the viral particles, specifically the nucleocapsid, in the nuclear membrane and inside the nucleus, but not the virus in the skin of long COVID mice (**C**–**E**). Scale bar = 35 µm. 4′,6-diamidino-2 phenylindole (DAPI [blue]), spike glycoprotein (green), nucleocapsid protein (red), and colocalization of (merged) nucleocapsid protein and spike glycoprotein around the nucleus as well as in the cytoplasm.

## Data Availability

The data presented in this study are available on request from the corresponding author. The data are not publicly available due to the University of Miami Miller School of Medicine’s privacy policy.

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
