# Peer review of "Dermatologic Changes in Experimental Model of Long COVID"

_microorganisms, 2024, doi:10.3390/microorganisms12020272_

Round 1
Reviewer 1 Report
Comments and Suggestions for Authors
Reduce the number of words in the abstract.
There is a lack of a group of mice treated only with SPIKENET that does not damage the dermal tissue.
MHV-1 infection assays
How many animals in total were used in the study?
It is necessary to ensure that the mice are infected (qPCR or FISH assays using specific regions of MHV-1)
Regarding the previous point, in what phase of the infection in Figure 1 did you take the skin samples? At the maximum peak of the disease, or before or after. Perhaps it is necessary to follow the viremia with qPCR in your model and choose the best moment for the study.
Long-COVID trials
One of the characteristics of Long-COVID is the absence of infection. How did they guarantee that after 12 months, this no longer existed?
If the skin regions of the samples were homogeneous, perhaps it would be worth making measurements of the adipose tissue, hair follicles, epidermal layer, dermal layer, etc., to make the study a little more quantitative.
Do mice develop clinical signs of Long-MHV-1?
Author Response
Response to Reviewer #1 comments:
- Reduce the number of words in the abstract.
We now reduced the number of words in the abstract (previously 355 and now 285).
- There is a lack of a group of mice treated only with SPIKENET that does not damage the dermal tissue.
Thanks for the suggestion. We dedicated a specific group for SPK alone, which showed identical picture to the healthy group (control). We therefore thought that adding another image which is identical to the healthy animals (controls) will not add any new information. So, we omitted SPK alone images (both in acute and long-term post infection) to avoid duplicating similarity to the control. Nevertheless, if the reviewer thinks that it’s necessary to add such an image, we will be happy to provide it. Moreover, for clarity we now add a sentence in the m/s elaborating SPK alone group with number of animals performed (i.e., has no effect in altering the skin changes) in the revised m/s.
- MHV-1 infection assays: It is necessary to ensure that the mice are infected (qPCR or FISH assays using specific regions of MHV-1). Regarding the previous point, in what phase of the infection in Figure 1 did you take the skin samples? At the maximum peak of the disease, or before or after. Perhaps it is necessary to follow the viremia with qPCR in your model and choose the best moment for the study.
If we understand the reviewers’ concern correctly, there are several references that demonstrate the viral load in MHV-1 infection, including our more recent studies which demonstrate organ specific viral presence. We now also show the presence of viruses in the skin post-acute infection (7 days post-infection-see new images in the revised m/s) and viral particles in long-term post-infection (12 months post-infection). The 7 days infection is identical to that in humans infected with the SARS-CoV-2. However, there are no clear data and specific time points for long-term effect in humans although we performed one-year post-infection. The references below refer to further details regarding MHV-1 infection. We now added these details and references in the method section of the revised m/s for more clarity.
- De Albuquerque N, Baig E, Ma X, Zhang J, He W, Rowe A, Habal M, Liu M, Shalev I, Downey GP, Gorczynski R, Butany J, Leibowitz J, Weiss SR, McGilvray ID, Phillips MJ, Fish EN, Levy GA. Murine hepatitis virus strain 1 produces a clinically relevant model of severe acute respiratory syndrome in A/J mice. J Virol. 2006 Nov;80(21):10382-94.
- Tian J, Kaufman DL. The GABA and GABA-Receptor System in Inflammation, Anti-Tumor Immune Responses, and COVID-19. Biomedicines. 2023 Jan 18;11(2):254.
- Gong HH, Worley MJ, Carver KA, Goldstein DR, Deng JC. Neutrophils drive pulmonary vascular leakage in MHV-1 infection of susceptible A/J mice. Front Immunol. 2023 Jan 6;13:1089064.
- Archer SL, Dasgupta A, Chen KH, Wu D, Baid K, Mamatis JE, Gonzalez V, Read A, Bentley RE, Martin AY, Mewburn JD, Dunham-Snary KJ, Evans GA, Levy G, Jones O, Al-Qazazi R, Ring B, Alizadeh E, Hindmarch CC, Rossi J, Lima PD, Falzarano D, Banerjee A, Colpitts CC. SARS-CoV-2 mitochondriopathy in COVID-19 pneumonia exacerbates hypoxemia. Redox Biol. 2022 Dec;58:102508..
- Cox G, Gonzalez AJ, Ijezie EC, Rodriguez A, Miller CR, Van Leuven JT, Miura TA. Priming With Rhinovirus Protects Mice Against a Lethal Pulmonary Coronavirus Infection. Front Immunol. 2022 May 30;13:886611.
- Pathak L, Gayan S, Pal B, Talukdar J, Bhuyan S, Sandhya S, Yeger H, Baishya D, Das B. Coronavirus Activates an Altruistic Stem Cell-Mediated Defense Mechanism that Reactivates Dormant Tuberculosis: Implications in Coronavirus Disease 2019 Pandemic. Am J Pathol. 2021 Jul;191(7):1255-1268.
- Zorzitto J, Galligan CL, Ueng JJ, Fish EN. Characterization of the antiviral effects of interferon-alpha against a SARS-like coronoavirus infection in vitro. Cell Res. 2006 Feb;16(2):220-9.
- Khanolkar A, Hartwig SM, Haag BA, Meyerholz DK, Harty JT, Varga SM. Toll-like receptor 4 deficiency increases disease and mortality after mouse hepatitis virus type 1 infection of susceptible C3H mice. J Virol. 2009 Sep;83(17):8946-56.
- Leibowitz JL, Srinivasa R, Williamson ST, Chua MM, Liu M, Wu S, Kang H, Ma XZ, Zhang J, Shalev I, Smith R, Phillips MJ, Levy GA, Weiss SR. Genetic determinants of mouse hepatitis virus strain 1 pneumovirulence. J Virol. 2010 Sep;84(18):9278-91.
- Ma XZ, Bartczak A, Zhang J, Khattar R, Chen L, Liu MF, Edwards A, Levy G, McGilvray ID. Proteasome inhibition in vivo promotes survival in a lethal murine model of severe acute respiratory syndrome. J Virol. 2010 Dec;84(23):12419-28.
- Kebaabetswe LP, Haick AK, Miura TA. Differentiated phenotypes of primary murine alveolar epithelial cells and their susceptibility to infection by respiratory viruses. Virus Res. 2013 Aug;175(2):110-9.
- Hua X, Vijay R, Channappanavar R, Athmer J, Meyerholz DK, Pagedar N, Tilley S, Perlman S. Nasal priming by a murine coronavirus provides protective immunity against lethal heterologous virus pneumonia. JCI Insight. 2018 Jun 7;3(11):e99025.
- Paidas MJ, Mohamed AB, Norenberg MD, Saad A, Barry AF, Colon C, Kenyon NS, Jayakumar AR. Multi-Organ Histopathological Changes in a Mouse Hepatitis Virus Model of COVID-19. Viruses. 2021 Aug 27;13(9):1703. doi: 10.3390/v13091703. PMID: 34578284; PMCID: PMC8473123.
- Paidas MJ, Cosio DS, Ali S, Kenyon NS, Jayakumar AR. Long-Term Sequelae of COVID-19 in Experimental Mice. Mol Neurobiol. 2022 Oct;59(10):5970-5986. doi: 10.1007/s12035-022-02932-1. Epub 2022 Jul 13. PMID: 35831558; PMCID: PMC9281331.
- Paidas MJ, Sampath N, Schindler EA, Cosio DS, Ndubizu CO, Shamaladevi N, Kwal J, Rodriguez S, Ahmad A, Kenyon NS, Jayakumar AR. Mechanism of Multi-Organ Injury in Experimental COVID-19 and Its Inhibition by a Small Molecule Peptide. Front Pharmacol. 2022 May 30;13:864798.
- Caldera-Crespo LA, Paidas MJ, Roy S, Schulman CI, Kenyon NS, Daunert S, Jayakumar AR. Experimental Models of COVID-19. Front Cell Infect Microbiol. 2022 Jan 5;11:792584.
- Ramamoorthy R, Hussain H, Ravelo N, Sriramajayam K, Di Gregorio DM, Paulrasu K, Chen P, Young K, Masciarella AD, Jayakumar AR, Paidas MJ. Kidney Damage in Long COVID: Studies in Experimental Mice. Biology (Basel). 2023 Jul 30;12(8):1070. doi: 10.3390/biology12081070. PMID: 37626956; PMCID: PMC10452084.
4, How many animals in total were used in the study?
We now provide the following details in the revised m/s. In the acute investigation, these mice were divided into 4 groups: (1) MHV-1 infection alone (n = 16), healthy control (n = 7), and SPK treated mice (n = 5). For the long-COVID study, we investigated a total of 12 mice (4 MHV-1 infection, 4 healthy control, and 4 SPK treated group). We also had SPK alone group both in acute and long-term studies (4 in acute, and 6 in long-term groups).
- Long-COVID trials: One of the characteristics of Long-COVID is the absence of infection. How did they guarantee that after 12 months, this no longer existed?
Please see just above our response to comment #3. Additionally, we recently reported the presence of SARS-CoV-2 in the brain of COVID-19 patient (1 year post-infection), and since studies have shown cell injury when exposed to viral particles (e.g., S1 protein), as well as we now show the presence of viruses and viral particles (new figure 10, only viral particles in the case of long-term post-infection), it is possible that the presence of viral particles for long-term may pose injury to the cells and organs including in the skin. We now add all these details in the revised m/s.
- If the skin regions of the samples were homogeneous, perhaps it would be worth making measurements of the adipose tissue, hair follicles, epidermal layer, dermal layer, etc., to make the study a little more quantitative.
We appreciate the reviewer’s comment. We measured the skin layers changes in and we now add the quantitation of skin thickness in the revised manuscript (see new figures 8 and 9).
- Do mice develop clinical signs of Long-MHV-1?
Please see above our response to comment #3 and 5.
Reviewer 2 Report
Comments and Suggestions for Authors
Main Research Question:
The main question addressed by the research is: What are the acute and long-term dermatologic changes post-COVID-19 infection, and can they be mitigated or reversed through treatment with a synthetic peptide, SPIKENET (SPK)?
Originality and Relevance:
The topic is highly original and relevant in the field. It addresses a specific gap by focusing on long-term dermatologic changes post-COVID-19, an area that has not been extensively explored. The use of an experimental model (MHV-1 in mice) to simulate SARS-CoV-2 infection effects adds novelty.
Contribution to the Subject Area:
This study adds significantly to the existing literature by providing detailed insights into both acute and long-term skin changes following COVID-19, highlighting specific alterations like hair follicle loss, collagen deposition, and epidermal layer devastation. The potential therapeutic effect of SPK is a novel addition, offering a potential avenue for treatment.
Methodological Improvements and Controls:
Major comments:
Mechanism of Action: The study could delve deeper into the mechanism of action of SPK, providing a clearer understanding of how it influences skin changes at the molecular level.
The study does not compare the effects of SPK with other potential treatments or a placebo. Including these controls would strengthen the conclusions regarding the specific efficacy of SPK.
Minor -Comments on Tables and Figures:
The figures, such as the histological images, effectively illustrate the skin changes in different phases of the study. They are well-labeled and support the textual findings.
Additional quantitative data in tables could enhance the clarity of the results, such as measurements of epidermal thickness.
Consistency of Conclusions:
The conclusions are consistent with the evidence and arguments presented, directly addressing the main research question. The findings on skin changes and the potential efficacy of SPK are well-supported by the experimental data.
Overall, the study provides valuable insights into the dermatologic impacts of COVID-19 and proposes a novel therapeutic approach, but the authors should consider improving the Conclusion section with a future perspective.
Appropriateness of References:
The references appear appropriate, providing context and supporting the research's relevance and methodologies. They are up-to-date, relevant to the topic, and include pertinent previous studies.
Comments on the Quality of English Language
Minor editing of English language required.
Author Response
Response to Reviewer #2 comments:
- The study could delve deeper into the mechanism of action of SPK, providing a clearer understanding of how it influences skin changes at the molecular level.
This is an important and intriguing question. While SPK protected the skin from both acute and long-term infection, it is unclear the precise mechanisms by which SPK prevented the skin defect. However, since the peptide was developed to specifically bind with the receptor binding domain of the S1 protein thereby it inhibits viral entry, and our reported studies also demonstrates its anti-inflammatory effect (check our references, 14-18), it is possible that the peptide may have prevented the skin defect by these 2 potential mechanisms. We now added a sentence regarding this aspect in the revised m/s.
- The study does not compare the effects of SPK with other potential treatments or a placebo. Including these controls would strengthen the conclusions regarding the specific efficacy of SPK.
Thanks for the valuable suggestion. We will certainly consider it in the future. Currently, we are expanding our experimental studies and further testing our compound in part of our plan.
- Additional quantitative data in tables could enhance the clarity of the results, such as measurements of epidermal thickness.
This suggestion holds significant value, as reviewer 1 also suggested the same. Consequently, we have incorporated a graphical representation of the variations in the thickness of epidermal, dermal, and adipose tissue layers in the revised manuscript, as depicted in figures 8 and 9.
- Consistency of Conclusions: The authors should consider improving the Conclusion section with a future perspective.
We have now modified the conclusion to reflect the suggestion.
- Minor editing of English language required.
Done
Reviewer 3 Report
Comments and Suggestions for Authors
The skin manifestations of COVID in the acute phase and Long-COVID are analyzed. In the acute phase, a papular eruption, a purpuric form due to the effects of coagulation alterations induced by the virus and an acrocyanotic-like form on the toes can be observed. At a clinical level, a hyperplasia of the sebaceous glands with an increase in hair follicles in the anagen phase is observed in the acute phase, while in Long-COVID skin atrophy and a significant reduction of hair follicles are observed.
At a clinical-histological level, in mice infected with MHV-1 a reduction in hair follicles is observed and treatment with the synthetic peptide SPIKENET (SPK) showed a substantial improvement in skin changes and in particular the restoration of morphological alterations and the number of hair follicles. It is hypothesized that the virus may elicit increased secretion of transforming growth factor beta (TGF-β), and that SPK may be evaluated as a potential effective drug in COVID infections.
The work is interesting, innovative and in my opinion it does not require changes.
Author Response
Response to Reviewer #3 comments: Thanks for the valuable appreciation on our m/s and for her/his approval for the publication.
Round 2
Reviewer 1 Report
Comments and Suggestions for Authors
The authors have covered all my doubts, so I believe the article can be published. I suggest authors reduce the abstract to a maximum of 200 words, as indicated by the publisher.
Reviewer 2 Report
Comments and Suggestions for Authors
Agree